# The Effect of Dietary Patterns on Inflammatory Biomarkers in Adults with Type 2 Diabetes Mellitus: A Systematic Review and Meta-Analysis of Randomized Controlled Trials

**DOI:** 10.3390/nu14214577

**Published:** 2022-10-31

**Authors:** Abril I. Sánchez-Rosales, Ana L. Guadarrama-López, Laura S. Gaona-Valle, Beatriz E. Martínez-Carrillo, Roxana Valdés-Ramos

**Affiliations:** 1Faculty of Medicine, Universidad Autónoma del Estado de México, Toluca 50180, Mexico; 2Multidisciplinary Clinic of Health, Universidad Autónoma del Estado de México, Toluca 50180, Mexico; 3Centro Médico Lic. Adolfo López Mateos, Instituto de Salud del Estado de México (ISEM), Toluca 50010, Mexico

**Keywords:** type 2 diabetes mellitus, chronic inflammation, biomarkers of inflammation, dietary patterns, Mediterranean Diet

## Abstract

Some evidence supports the fact that chronic low-grade inflammation contributes to the physiopathology of type 2 diabetes mellitus (T2DM), and circulating markers of inflammation (e.g., C-reactive protein (CRP), pro- and anti-inflammatory biomarkers (e.g., adiponectin), and endothelial function markers could indicate an ongoing pathology. Following certain dietary patterns (DPs) may result in favorable changes in inflammatory biomarkers. The overarching aim of this systematic review and meta-analysis is to explore the inflammatory effect of healthy DPs on inflammatory biomarkers in adults with T2DM. A systematic search of the literature was conducted using the electronic databases MEDLINE, SCOPUS, and Cochrane Central Register of Controlled Trials. A total of 10 randomized controlled clinical trials (RCTs) were analyzed. In our linear meta-analysis, the random-effects model was applied to estimate standardized mean differences (SMD) to associate the effect of the interventions. Dietary Approaches to Stop Hypertension (DASH), Diabetes UK healthy eating, Mediterranean Diet (MD), Diabetes Prevention Program (DPP), and the American Heart Association’s Therapeutic Lifestyle Changes diet were associated with a significant reduction in CRP (SMD: −0.83, 99% CI −1.49, −0.17, *p* < 0.001; I^2^ 94%), while plasma levels of adiponectin were significantly higher with the intake of MD, DPP, and Diabetes UK healthy eating (SMD: 0.81, 99% CI 0.06,1.56, *p* < 0.005; I^2^ 96%), both of which indicate less inflammation. Sensitivity analyses were carried out, and potential publication bias was examined. In conclusion, low- moderate-quality evidence from RCTs suggests that, for the DPs evaluated, there are favorable changes in CRP and adiponectin.

## 1. Introduction

The prevalence of type 2 diabetes mellitus (T2DM) is increasing at alarming rates globally [1]. By 2045, 629 million people are expected to be diagnosed with T2DM [2]. There is some evidence supporting the fact that chronic low-grade inflammation contributes to the physiopathology and progression of T2DM. Circulating markers of inflammation (e.g., C-reactive protein (CRP), pro- and anti- inflammatory biomarkers (e.g., adiponectin), and endothelial function markers could indicate an ongoing pathology (i.e., higher levels of CRP, interleukin-6 (IL-6), TNF-α and lower levels of adiponectin) [3]. While limited research on this topic supports the association between circulating markers of inflammation and pathology, this area of research shows interesting research potential [4]. For this reason, the few available studies must provide a reference framework to close gaps about chronic low-grade inflammation on T2DM and to analyze under which conditions of participants need to be (e.g., glycemic control) for these effects can be achieved [5].

The assessment of diet is a complex issue because people eat meals instead of separate foods or nutrients; for this reason, it has been proposed to evaluate dietary patterns (DPs) and to obtain a whole scope of diet [6,7]. Currently, the relationship of the effects of a dietary pattern (DP) on inflammatory biomarkers is through properties that modulate their molecular interactions and participate in signaling and inflammatory pathways, providing a defense and regulating their intermediaries in the immune system [8,9]. It is hypothesized that consumption of anti-inflammatory components (i.e., antioxidants, prebiotics, and mono and polyunsaturated fats) provided by vegetables, fruits, wine, seeds, oils, white meat, etcetera, and foods with bioactive compounds (i.e., whole grains, oil, seeds, coffee, and alcohol) results in favorable changes in inflammatory biomarkers [10].

Every pattern reflects different combinations of food intake and could contribute to a positive balance in inflammation, therefore, improving glycemic control, due to mechanisms of propagation of inflammation in adipose tissue and skeletal muscle (molecular nutrient sensors and anti-inflammatory pathways can promote insulin sensitivity), and having effects on regulating systematic insulin sensitivity [11,12]. There are very few reviews focusing on T2DM and the control of the inflammatory process through DPs that could eventually reduce chronic low-grade inflammation.

The overarching aim of this systematic review and meta-analysis is to explore the effect of utilizing DPs, commonly referred to as “healthy”, on inflammatory biomarkers, summarizing the available interventions targeting low-grade chronic inflammation and the impact of each pattern on parameters that describe inflammation, as this will help identify relevant gaps in knowledge about the variety of dietary interventions, T2DM, and the use of biomarkers of inflammation.

## 2. Materials and Methods

In accordance with the Preferred Reporting Items for Systematic Reviews and Meta-Analyses guidelines (PRISMA) [13], this manuscript is intended to be accompanied by a systematic review (see Appendix A: PRISMA 2020 Checklist). The protocol was developed a priori and registered with the National Institute for Health Research’s International prospective register of systematic reviews (PROSPERO).). ID: CRD42021246854.

### 2.1. Search Strategy

For data collection, a systematic search of the literature was conducted using the electronic databases MEDLINE, SCOPUS, and Cochrane Central Register of Controlled Trials until 22 January 2022, with restrictions to Randomized Clinical Trials (RCTs). The studies used were published between 2010 and 2021, a date range selected by the current criteria of lag time over the course of a decade, and the articles were confined to those published in the English language. The search strategy was peer-reviewed by R.V.R. and B.M.C., with disagreements resolved by consensus. Search terms were determined according to the Scientometrics Laboratory of Information and Biological Informatics. We utilized tools such as HONselect to define MeSH^®^, and words and/or their combinations were searched using Bolean operators (e.g., ‘AND’ and ‘OR’).

Examples of search terms included were: (“dietary pattern “OR “diet* pattern” AND (“adult”) in combination with (“biomarker” OR “inflammation” OR “inflammatory”) AND (“diabetes mellitus type 2” OR “type 2 diabetes mellitus”) (see the Appendix A for the list of search terms).

Using the suggested algorithm, we searched with keywords (presented in the title and abstract), combinations and limits were included (humans, adults, and English language) and incorporated into the reference lists, and these articles were also screened.

### 2.2. Eligibility Criteria

We selected studies in accordance with our pre-specified clinical research question and the Population, Intervention, Comparator, Outcomes, Study Design (PICOS) method. We did not consider minimum requirements for sample size (sensitivity analyses should be carried out accordingly to analyze the sample). Considering the eligibility criteria, if more than one article reported on the outcomes of one study, the longest follow-up period was used. PICOS criteria for inclusion of studies were used to select studies of inclusion: adults with T2DM (≥ 18 years of age) in studies with *a priori* (i.e., pre-selected/designated patterns to be implemented); ‘Diet Pattern’, ‘Mediterranean Diet’, ‘Healthy Diet´, and ‘Anti-inflammatory Diet´ with a comparator of intervention versus control or usual diet; ‘Habitual Diet’ and ‘Control Diet´, which provided outcomes of biomarkers descriptive of chronic inflammation and records of change from baseline values (‘pro-inflammatory biomarkers´: CRP, TNF-α, IFN-γ, IL-1β, IL-6, visfatin, resistin, E-selectin, bacterial lipopolysaccharides (LPS), retinol-binding protein 4 (RBP4), or leptin; ´anti-inflammatory biomarkers´: adiponectin, ratio of adiponectin to leptin; and ‘endothelial function´: vascular cell adhesion molecule 1 (VCAM-1) and ICAM-1), with the following study design: RCTs covering parallel and crossover study designs, without minimum intervention follow-up.

Applicable exclusion criteria were considered: I. studies including pregnant, breastfeeding, or menopausal women; II. studies with an approach in renal disease, chronic kidney disease, kidney failure, kidney replacement therapy, or heart failure; II. symposium/conference/interview abstracts, scientific divulgation (science outreach), or study protocols/pilot study; and III. studies reporting post-prandial effects of DPs or post-prandial measures only if it was not possible to separate and analyze effects of DP change (e.g., a combination of measures, indexes), and exploratory patterns or a posteriori patterns (i.e., not pre-selected/designated patterns).

Studies reporting incomplete data on biomarkers of inflammation were excluded after the corresponding author was contacted for Appendix A and could not be retrieved.

### 2.3. Selection of Studies

Initially, potentially eligible articles were screened by title and abstract; if these were not available or did not give necessary information to obtain scope and purpose, we reviewed concerning inclusion criteria and the full texts of the articles to decide whether to include them.

In the case the record of change from baseline values were unavailable (i.e., studies of a single measure), post-intervention mean values were used as an alternative. Additionally, we considered as part of outcomes of interest (secondary outcomes); measures of glycemic control or monitoring of fasting glucose, glycated hemoglobin (HbA1c), and insulin. Studies including secondary outcomes with blood levels of biomarkers could be eligible. In the case of studies reporting on the progress of a previously implemented intervention, the most recent version of the results was used and the study with the longest periods was included.

We defined “DP” with a standard definition used in the field: “DP is traditionally defined as an eating pattern made up of foods and beverages and not as a diet based on macronutrients or nutrients alone” [14]. We also considered including dietary portfolios and eating guidelines as part of a model or guide that is indicative of quantity, variety, combinations, and frequency of various food [15].

Selection process was undertaken by A.I.S.-R. and R.V.-R. Any doubts in the selection and/or decision-making of studies were dissolved by an independent author (B.E.M.-C.).

### 2.4. Data Extraction

Articles were saved in the Mendeley reference manager software, desktop 1.19.4 (copyright © 2020 Mendeley Ltd, Amsterdam, NL.), and duplicate articles were eliminated manually. When papers did not match the inclusion criteria, the main reason for inclusion was recorded. Afterwards, a list of studies was designed. We built a database and recorded the following data: year of publication, author(s), study design, main objective, reasons to either include or exclude each article, study sample and size (including mean age, age range, number of women and men in each study, gender proportions, and mean of body mass index), study duration, details of the intervention (location, recruitment) and principal outcomes of interest measured (levels of biomarkers measured with laboratory technique), dietary assessment tools, DP designated by the authors, energy content of diets, descriptions of intervention and control groups, statistical analysis including statistical results, control variables, key findings, main conclusions, important aspects about discussion, ethical declarations, financial and conflict of interest statements, and relevant declarations by the authors (e.g., biases and limitations). For studies where required data were not available, the corresponding author was contacted for Appendix A. Outcomes between intervention and control groups were clustered by inflammatory biomarkers separately in accordance with the Cochrane Handbook for Systematic Reviews of Interventions [16]: when methodology from studies implicates more than one intervention group gathering inclusion criteria.

Material extraction was conducted by one author (A.I.S.-R.) and presented advances to our group study. Disagreements were discussed to reach a consensus.

### 2.5. Quality Assessment and Risk of Bias of the Included Studies

We used system GRADE criteria (Grading of Recommendations, Assessment, Development, and Evaluation) to assess certainty and their Guideline Development Tool to create a format using GRADEPro©. We elaborated summary tables of findings and included PICOS criteria. In the process of downgrading criteria, we scored according to the following: (a) study limitations: risk of bias (publication bias), (b) indirectness, (c) inconsistency (heterogeneity), and (d) imprecision.

Risk of Bias from the studies was assessed by “the risk of bias assessment tool” from the Cochrane Collaboration RoB 2.0. This tool comprises the following designated criteria of bias domain: (a) selection, (b) detection/performance, (c) attrition, and (d) reporting bias. Each bias domain was classified as “low risk”, “high risk”, or “unclear risk”. In the domain selection and performance/detection, we decided to consider studies with double blinding (considered between control and experimental group and personnel) as “low risk”. To determine the potential risk of publication bias, we constructed funnel plots and their symmetry within mean differences, their standard errors were plotted, and the presence of publication bias was determined.

We applied independent processes to assess Risk of Bias, whereas quality assessment was independently performed by both authors (A.I.S.-R. and R.V.-R.). In the assessment of certainty and Risk of Bias every author was able to review and verify the final decision-making. Discrepancies were resolved in consensus.

### 2.6. Data Synthesis and Analysis

We conducted data analysis using the statistical measure reported in each study (i.e., means, mean differences, min, quartile 1, median, quartile 3, and max). In the case of studies outcomes, we compared inconsistent results between studies with different cut-off points (e.g., CRP ≥ 3 mg/L), which were reported in their unit of measurement. Data synthesis of all selected studies was identified, and each DP was labeled with key terms (i.e., the country of the study research or the name of the study). Quality synthesis, to compare type of DP involved, type of control diet, and biomarkers, was measured. We grouped for similar characteristics and composition of general DPs: we considered classical approaches with respect to dietary interventions in patients with diabetes.

#### 2.6.1. Meta-Analysis

A meta-analysis was conducted in support of outcome parameters to find out the associated intervention effect between basal and final interventions. We considered mean differences of the DP and control groups; in addition, classification of parameters (pro and anti-inflammatory biomarkers) was required for the analysis, and we decided not to mix biomarkers. If >3 or more articles reported the same biomarker, they were pooled for meta-analysis. We utilized median values and interquartile ranges, and when there was no statistical significative modification between values it was reported, absolute final intervention values were used to improve clinical translation of the pooled estimates. If none of these strategies was successful, methods for an estimated mean of the sample [17,18,19] were used. In the case of standard deviations were not provided (after contacting the authors) estimated standard deviations of the sample [20] were utilized.

The random-effects model was applied to estimate standardized mean differences (SMD) of continuous data. For each study outcome represented in the main analysis, we compared changes between intervention and control diets (longest follow-up mean versus basal mean and their standard deviations). To point out each specific effect size, forest plots were created with a % CI. Effect sizes are presented as SMD, 99%CI; findings were considered statistically significant if the 99%CI did not cross the zero-point estimate line and *p* < 0.01. We backed up the heterogeneity of the outcomes using the I^2^ test, and we considered levels of 25% to be low, 50% to be moderate, and 75% to be high.

All information was examined on Review Manager software version 5.4.1 by The Cochrane Collaboration©, Copenhagen, Denmark.

#### 2.6.2. Sensitivity Analyses

For modifications in CRP (mg/L) between DP and control groups, sensitivity analysis was done. When multiple pro-inflammatory biomarkers were evaluated in the same paper, CRP was prioritized for its function as a traditional inflammatory biomarker, and we implemented converting between high- and low-sensitivity CRP that allows comparisons calculated by formula (full details have been published elsewhere [21,22]). We established in the PROSPERO protocol registration, a subgroup analyses guided by kind of DP, we hypothesized that a subgroup of DP with similar characteristics obtain different results.

Univariate meta-regression analyses were not performed as procedures require at least ten studies (studies had insufficient information).

## 3. Results

### 3.1. Description of the Included Studies

Our literature search found 842 references from a systematic search of electronic databases, of which 47 full-text-available studies were retrieved and screened for eligibility as part of the selection process (Figure 1). After reviewing the full texts of 19 articles, nine were eliminated because they did not meet the inclusion criteria; one study was duplicate study population [23]; two studies had ineligible study design, and they were excluded because they reported the outcome for adherence to the Mediterranean diet (MD) as a binary variable (adherence or not adherence) without quantitative biomarkers [24,25]; two studies were part of a macro-project, and we used the eligible study version with the longest duration period [26,27]; one study compared lifestyle interventions with the use of insulin as medication [28]; one study showed incomplete monitoring assessment (no final values were presented, only basal values) [29]; one was a pilot study [30]; another was excluded due to a lack of data about biomarkers which we could not obtain even after contacting the authors (only mentioned their evaluation but did not report it) [31]. Finally, a total of 10 clinical trials [32,33,34,35,36,37,38,39,40,41] met the inclusion criteria and were included in our systematic review.

The articles analyzed in this review were published between 2011 and 2018. The study countries were the United Kingdom, Italy, Spain, Australia, the USA, Mexico, Israel, Iran, and Sweden. Studies were RCTs with a duration of up to 8.1 years and enrolled a total of 2992 subjects. The number of patients by gender was male, 1430 (47.80%), and female, 1562 (52.20%).

We found the following healthy DPs: Diabetes UK healthy eating guidelines [35], American Heart Association´s Therapeutic Lifestyle Changes diet [38], Anti-inflammatory dietary portfolio [40], Dietary Approaches to Stop Hypertension (DASH diet) [39], The Diabetes Prevention Program (DPP) [26], and the Mediterranean Diet (MD) [32,33,34,37,41]. The control diet was a low-fat diet [32,33,34,37,41], habitual diet [35], or prescribed diet [36,38,39,40]. An overview of publications included in the systematic review is shown in Table 1.

### 3.2. Risk of Bias of Included Studies and Quality of Evidence

Assessment of Risk of Bias was evaluated in the studies, and objective rating (i.e., low risk”, “high risk”, or “unclear risk) was assigned to each publication by type of domain. Studies with the most positive criteria were PREDIMED (five domains obtained “low risk” of eight criteria) [34] and LOOK AHEAD study (six domains obtained “low risk” of eight criteria) [36], rating of bias domains for every study is presented in Figure 2.

Other biases we found in the studies were (i) statistical analyses, (ii) conflicts of interest, (iii) follow-up, (iv) equilibrium in the arms, (v) enrollment, and (vi) study design/treatment (see Appendix A. The “Risk of Bias” tool). The type of bias was evaluated of the 10 studies included 100% had a low risk in random sequence generation, 75% had low risk in allocation concealment. Around 25% of studies used blinding of participants and personnel, and just half blinded the outcome assessment (Figure 3).

Funnel plots were generated for outcome measures provided by five different trials [32,33,34,35,36,37] for adiponectin and seven studies for CRP [32,33,34,35,36,37,38] Adiponectin´s plot showed little to moderate asymmetry, indicating that publication bias cannot be entirely excluded as an aspect that impacted the present meta-analysis (see Appendix A; Funnel plot of comparison: final values, outcome). CRP´s plot shows little asymmetry; nevertheless, an explanation for this could be those small studies providing inconclusive or unsuccessfully data have not been published (see Appendix A Appendix A; Funnel plot of comparison: final values, outcome).

Certainty and quality of evidence were rated, adiponectin obtained a low certainty and CRP a moderate certainty (Table 2).

### 3.3. Primary Outcomes: Inflammatory Biomarkers and Dietary Patterns

All studies reported at least one parameter using the following biomarkers of inflammation: TNF-α [34], IL-6 [34,35,37], visfatin [34], resistin [34,38], E-selectin [38], LPS [40], leptin [34,41] VCAM-1 [38], and ICAM-1 [34,35,38].

Maiorino et al. [32] and Lasa et al. [34] found that IL-6 and ICAM-1 levels were statistically significantly decreased, and adiponectin levels were statistically significantly increased in most trial years duration, while in the control diet, there was no significative changes (Table 1 Summary of studies included). In the Look AHEAD study [36], they found that the experimental group showed significant improvements. Adiponectin increased significantly with the experimental diet compared with controls in both men and women, and decreases in CRP (−43,6%, *p* < 0.001) were also observed. Fernemark et al. [41], in which leptin levels decreased. In the ACTID study [35] after 12 months, mean of CRP levels were substantially attenuated; with a reduction of relative risk of 0.72 (0.55–0.95).

We clustered studies by results and similar features, which were summarized according to biomarkers of inflammation (adiponectin, IL-6, CRP, leptin, and I-CAM) utilized more in RCTs, and we grouped each DP. Qualitative synthesis is presented in Table 3. The cluster “Cardio Protective” included recommendations by the American Heart Association and DASH diet [42]. The cluster “Specialized diet for subjects with diabetes” included DPP and the Diabetes UK healthy eating guidelines [43].

The DP most used was the MD [32,33,34,37,41], and we described and compared the food composition of different styles of MD among studies (see Appendix A. Characteristics and properties of components of Mediterranean Diet styles).

#### 3.3.1. Secondary Glycemic Control Outcomes

Studies assessed glycemic control at baseline through levels of fasting glucose and HbA1c. Insulin resistance was evaluated by Homeostatic Model Assessment of Insulin Resistance (HOMA-IR). Some studies reported changes in mean differences. In the MEDITA trial [32] using MD, changes in the HOMA-IR (3.3 ± 1.2, *p* = 0.01) were reported; in the PREDIMED study [34] fasting glucose in MD was –4.2 (37.3) mg/dL, *p* < 0.001; and the DASH diet [39] reported, in the final intervention, levels of fasting glucose of −13.9 ± 4.5 mg/dL, *p* < 0.05. Itsiopoulos et al. [37] reported a percent change in HbA1C of 6.8% (6.3–7.3), *p* = 0.12, with respect to the control diet, and levels of fasting glucose of 8.9 (mmol/L) (7.8 –10.0), *p* = 0.276. In the Look AHEAD study [36], changes in HbA1c % of −0.7 ± 1.0, *p* < 0.001, and fasting glucose (mg/dL) of −21.7 ± 44.4, *p* < 0.001, were reported. Table 1 shows a summary of the studies included for all the glycemic control outcomes.

#### 3.3.2. Meta-Analysis

A total of eight RCTs were included in the meta-analysis. In our linear analysis, we included and analyzed outcome measures provided by five different trials [32,33,34,35,36] for adiponectin and seven studies for CRP [32,33,35,36,37,38,39]. The biomarkers TNF-α [34], IL-6 [34,35,37], visfatin [34], resistin [34,38], E-selectin [38], LPS [40], leptin [34,41], VCAM-1 [38], and ICAM-1 [34,35,38] were not included because there were not enough studies and available data to compare. Studies that did not provide sufficient data (intervention and control or comparison) were not analyzed in the mean difference of our meta-analysis [40,41].

The pool estimated of all studies combined was associated with the changes calculated. The healthy DPs included were DASH [39], Diabetes UK healthy eating [35], MD [32,33,37], DPP [36], and American Heart Association´s Therapeutic Lifestyle Changes [38] and were associated with a significant reduction in CRP (SMD: −0.83 mg/L, 99% CI −1.49, −0.17, *p* < 0.001; I^2^ 94%%), while plasma levels of adiponectin were significantly higher with the healthy DPs included: MD [32,33,34], Diabetes UK healthy eating [35], and DPP [36] (SMD: 0.81 μg/mL, 99% CI 0.06,1.56, *p* < 0.005; I^2^ 96%). The Results of the meta-analysis are presented in Figure 4**.**

#### 3.3.3. Sensitivity Analyses/Subgroup Analyses

Subgroup analyses were performed on MD since three or more studies had data available for sensitivity analyses (Figure 5). The effect adiponectin levels in sub-groups by type of diet (MD) was associated with an increase in adiponectin levels (SMD 0.88 μg/mL, CI 0.14, 1.62, *p* = 0.002; Z = 3.06) and a reduction in CPR levels (SMD −0.37, 99% CI −1.37, 0.64, *p* = 0.35, Z = 0.94). The analysis for the other subgroups is presented in the Appendix A.

## 4. Discussion

The aim of this study was to identify RCTs on inflammation and DPs in T2DM by considering different approaches and by summarizing evidence via a meta-analysis. To our knowledge, this is a pioneer systematic review on the topic; previous systematic reviews have centered exclusively on glycemic control and *a priori* [44] or exploratory DPs [45] or did not separately focus on different dietary approaches to explore DPs and thus to summarize the evidence in T2DM and balance of inflammatory processes [46,47,48,49,50,51].

### 4.1. Summary Findings

In this systematic review and meta-analysis, we found six different healthy DPs, and their effect on eleven parameters of inflammation. In the systematic review, we evaluated ten RCTs, and we analyze, in our linear meta-analysis, the changes in values of CRP and adiponectin from eight studies.

### 4.2. Hypothesis/Reasons of Our Findings

The analysis of RCTs included in our study supports the theory that DPs could impact on biomarkers of inflammation and, therefore, values of glycemic control. Food composition of the DPs and duration of the interventions could be reasons for these effects [52,53].

We think that the outcomes in our study could be explained by basal measurements of control glycemic. A systematic literature review suggests that it is necessary to focus on the initial state of glycemic control [54]. With respect to glucose and according to the values of glycemic control of the American Diabetes Association (ADA) [55], only one study had participants with adequate glycemic control in basal measurements [37]. Concerning glycemic control in experimental groups, seven studies [32,34,35,36,37,39,40] demonstrated beneficial changes in values after experimental DPs. In the PREDIMED study [34], a favorable change in fasting glucose was observed (compared versus control group −3.6 mg/L, *p* < 0.01), which is remarkable because in the intervention of PREDIMED study, participants received intensive education to follow MD with recommendations and quantitative aspects. Additionally, also with MD, Itsiopoulos et al. [37] observed clinically and statistically significant falls in HbA1c, which reinforced the idea that MD provides benefits on glycemic control. The DASH diet [39] also reported, in the final intervention, lower levels of fasting glucose (−13.9 (4.5) mg/dL, *p* < 0.05), in a period of eight weeks. In this same period, Sauder et al. [42] found that there were no differences in values of glycemic control after an experimental diet. Furthermore, in the Look AHEAD study [36], DPP showed a 0.7% drop in A1C, resulting in a 43.6% decrease in median hs-CRP. The improvement in glycemic control achieved with DPP was also in one year. This evidence showed according to the ADA goals [56].

In our study, we found that healthy DPs focus mainly on the consumption of unsaturated fatty acids, whole grains, and high amounts of antioxidants and phytochemicals, in contrast with two studies included in the study [36,39] which mainly focus on glycemic index with low-carbohydrate content. Furthermore, based on our analyses which may influence the DPs studied and their classification, some authors proposed that the traditional treatment for patients with T2DM is a low-carbohydrate diet [50], although we found four studies that evaluated the low-fat diet as control diet [32,33,34,41]. A meta-analysis assessing the effect of different diets on markers of inflammation in patients with metabolic syndrome described a positive effect of low-fat diets on the reduction in CRP [57]. In contrast, Itsiopoulos et al. [37] administered a habitual diet in participants with adequate glycemic control. Sauder et al. [38] defined the same type of diet between groups; however, they added a functional component. Indeed, it is important to suggest an experimental diet that considers nutritional content (proportion of carbohydrates, protein, and type of fat) because inflammation plays a key role at all stages of the disease, based on pathogenesis and modifying related inflammatory pathways, as the visceral fat is directly associated with the production of pro-inflammatory mediators [58]. The principal fat in the studies was provided by mono-unsaturated fatty acids and polyunsaturated fatty acids, with restrictions in saturated fatty acids. Additionally, weight loss can reduce several pro-inflammatory markers such as CRP, IL-6, and TNF-α [59], and, in general, the glycemic state with the improvement of inflammation markers [25]. In the Look AHEAD study [36], a significant weight loss and improvement in levels of CRP (−1.24, *p* < 0.001) were reported, while the DASH diet [42] showed that a calculation of individual energy requirements could improve CRP levels (−2.04, *p* = 0.02). An experimental diet should be established in the studies of DPs in accordance with type of fat, weight loss, and type of control diet in T2DM (subgroups of low-fat versus low-carbohydrate diet) populations to represent a real-life clinical practice.

Some doses and components of healthy DPs are still controversial, and the exact doses of some compounds for T2DM are unknown. In our Systematic Review, we observed doses and amounts, and only dietary recommendations to achieve a DP. For example, the amount of alcohol is controversial. The Direct Trial [33] did not include it, and the other MD styles recommended moderate consumption of red wine [32,34,37,41]. The Dietary Inflammation Index has proposed it as an anti-inflammatory compound [60]; however, the World Health Organization disapproves of any consumption of alcohol [61].

It seems that the results of the studies indicate that dietary health recommendations also could be useful. A systematic review that provided information about dose–response reported that when applying only healthy dietary recommendations, statistically significant associations between MD and CRP were found; in contrast, when utilizing a structured diet and an olive oil diet, no associations with inflammatory markers were observed [62]. We also explored similar characteristics in an experimental diet, in which various amounts and doses modified the structure of the regimen and contributed to variations; however, the same DP outcomes resulted. For example, we found different compositions of MD in the studies. MD was used in five RCTs, some studies coming from the Mediterranean region [32,33,34] and others with different geographical locations [37,41]. Mainly, the characteristic foods of the MD that would benefit the degree of inflammation were used; we found that the usual diet of European populations, principally in Mediterranean countries, tends to include the consumption of seafood, vegetables, and fruits, whereas American populations mostly adhere to whole cereals, legumes, and white meat containing different loads of inflammatory foods [63].

To our knowledge, evidence suggests that components of healthy DPs stimulate anti-inflammatory actions [64] and could possibly be a practical treatment for T2DM [65]; however, based on the findings of similar studies [66], avenues for future research included doses and components of healthy DPs [67].

In line with the hypothesis, the duration of interventions could be impact in the results and mostly should be > six months to observe favorable changes in T2DM [68]. In our study, we found that the duration of the interventions was, on average, mostly one year. We believe that the DPs that we analyzed in our study may be effective dietary strategies for stimulating anti-inflammatory actions in T2DM that are used as a primary intervention in long-term consumption [69,70].

Outcomes could be affected by a high heterogeneity in the studies evaluated. In the present study, the data suggest that the I^2^ measure of heterogeneity increased with broadening regions and multicenter studies. The DPs are also a reflection of habits and environment [71]; we hypothesized that location also could interfere with our results. It is beyond the scope of this study, and necessary results may further support research into the application of healthy DPs for individuals with T2DM.

### 4.3. Similar Outcomes

The results of the Meta-analysis indicate that favorable changes in biomarkers of inflammation could be participate in the balance of inflammation. CRP a pro-inflammatory biomarker showed the pooled effect of experimental diet to reduce statistically significant levels of CRP (−0.83, −1.49 to −0.17), and adiponectin an anti-inflammatory biomarker showed the pooled effect of experimental diet to reduce statistically significant levels of adiponectin (0.81, 0.06 to 1.56). Similar evidence was found in a meta-analysis of 17 RCTs including 2300 subjects, supporting evidence that both CRP and adiponectin levels were favorably affected following interventions with healthy DPs [72]. Contrary to the main analysis, in our sub-group analysis by a kind of DP, we found a significant increase in adiponectin levels (0.88, 0.14 to 1.62), but no significant decrease in CRP (−1.37 to 0.64). Due to the lack of number of studies, interpreting this should be considered exploratory, and the results cannot confirm if CRP did not show lower levels using MD; one possible explanation is elevated inter- and intra-methodological variability among the studies even if was utilized the same control and experimental diet.

A profile of biomarkers to describe the process of chronic inflammation is still quite inconsistent, as we, do not currently know which cut-off points are for a specific population or which to use and this could be helpful to establish an inflammatory state in individuals.

We know that only two biomarkers are insufficient to describe the inflammation process; however, we found that CRP, adiponectin, IL-6, ICAM-1, and TNF- could be integrated into this profile, in accordance with the results in the previous a systematic literature reviews [73,74]. In our quality synthesis, we found the presence of TNF-α [34], IL-6 [34,35,37], visfatin [34], resistin [34,38], E-selectin [38], LPS [40], leptin [34,41] VCAM-1 [38], and ICAM-1 [34,35,38] among the studies; nevertheless, we could not include all of them in our meta-analysis due to lack of data and number of studies for comparison.

### 4.4. Implications for Further Research

There are diverse DPs implemented in the T2DM population with the purpose of modulating chronic inflammation. In a 2013 position statement, the American Diabetes Association suggested five eating patterns for the management of diabetes: MD, low-fat, low-carbohydrate, vegetarian and vegan, and DASH [56]. The results of this review provide more evidence about any DPs that may help decrease low-grade inflammation. We compared different characteristics of foods shared in MD, which could explain the total effect of interventions as most of the associations in the studies were linked with CRP; this also is supported in a systematic literature review of epidemiological studies [75]. There was an inverse association relating inflammatory biomarkers to vegetables, whole grains, fish, and fruits based on *a priori* healthy DP, particularly with CRP concentrations. Furthermore, it is well known that “high-quality carbohydrates” in the diet lead to a higher intake of antioxidants, magnesium, and fiber, which are linked to positive associations with adiponectin [76]. Some studies also supported this; for example, a systematic literature review concluded that MD and DASH diets were associated with a decrease in inflammatory markers [77]; this is in accordance with studies included in our review of healthy DPs as well. In contrast, a very recent meta-analysis and review of RCTs in various chronic diseases concluded that the MD, but not the DASH diet, reduced IL-1b, IL-6, and CRP [59]. In our meta-analysis, we found similar outcomes, which could explain the total effect of interventions, as most of the negative associations in the studies were linked with CRP and positive associations with adiponectin [78].

Further investigations should involve large trials that measure all these DPs in subjects with T2DM with similar characteristics and appropriate follow-up with the same control diet (e.g., habitual diet). Additionally, further research should aim to guarantee a similar DP intake. Approaches between insulin sensitivity and immunomodulatory effects of each dietary regimen or DP would be another area to explore; for example, we found that MD is a DP with effects in glucose and insulin levels and favorable changes in PCR and adiponectin levels. Additional research should also try to include several pro- and anti-inflammatory factors in the progress of T2DM. The profile of the pro-inflammatory status in T2DM and its nutritional treatment is still a field of opportunity. In clinical practice, there are many advantages when DPs are individualized to a particular region because they are easier to implement in everyday life.

### 4.5. Strengths and Limitations

The major study strengths rely on utilizing only RCTs. Studies were selected following a scrutiny process with well-defined inclusion criteria; furthermore, we used methods standardized for the recollection of nutritional, clinical, and biomarker information. Although there is no standard definition for the MD and recommendations are based on populations, we found overlapping elements to analyze.

Among the study limitations, we acknowledge that we gathered information on intermediate endpoints as secondary outcomes in the studies. Deserving consideration is the lack of standardization of inflammatory outcomes in research, this limited comparison between studies makes analysis difficult, which means adequate comparisons cannot be made in meta-analyses. Additionally, we could not include some biomarkers in our meta-analyses.

Another study limitation is that we observed a high heterogeneity. We believe the main reasons were the variety of foods in the DPs analyzed, thereby potentially affecting the effects of the meta-analysis. This could be related to the wide variability in results from data collection and analysis in the various RCTs, which are not uniformly adjusted.

## 5. Conclusions

This review reveals that, due to study characteristics and types of DPs, including quantitative variables and small size of this review along with disagreements across studies, it is difficult to generalize and make further conclusions.

We concluded that low- to moderate-quality evidence from RCTs suggests that healthy DPs were associated with favorable changes; the pool estimates of all studies combined were associated with a significant reduction in CRP, while plasma levels of adiponectin were significant higher, both of which indicate less inflammation. The findings from this study provide evidence to support the implementation of DPs for assisting in the better management of chronic inflammation in subjects with T2DM.

In the current context, intervention studies provide evidence to study DPs, and it is necessary to undertake studies that reflect usual dietary intake. Clinical trials are consequently required using the same control diet.

We propose to continue exploring healthy DPs and their effect on inflammatory biomarkers to evaluate changes over time in T2DM as prognostic risk factors, which should be included as part of the nutritional monitoring.

## Figures and Tables

**Figure 1 nutrients-14-04577-f001:**
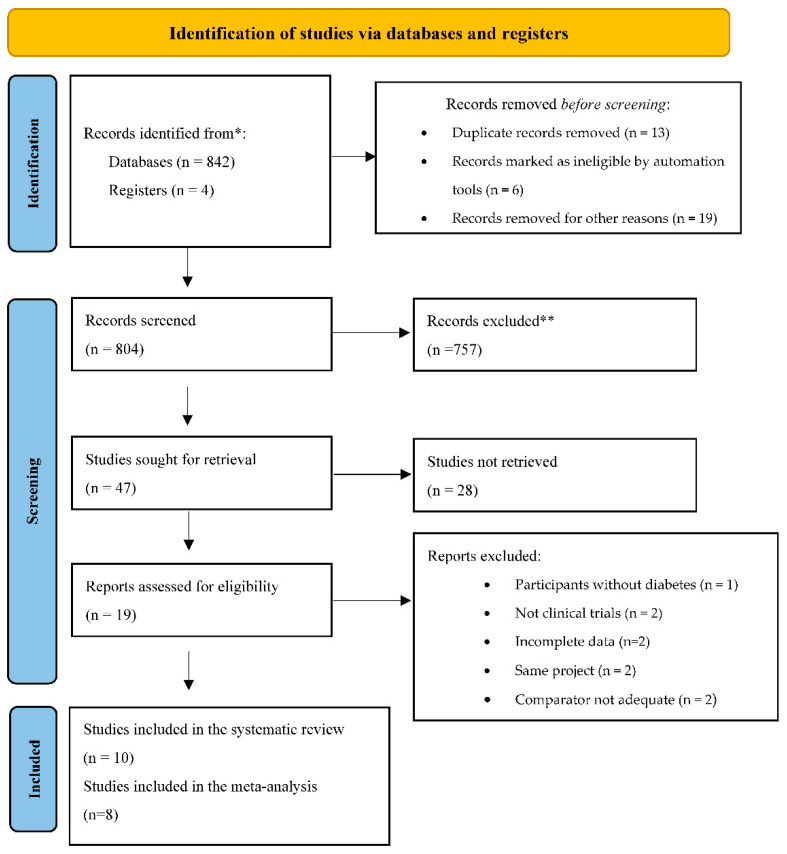
PRISMA Flow Diagram. PRISMA 2020 flow diagram for new systematic reviews that include searches of databases and registers only. * Databases: MEDLINE (589), SCOPUS (136), COCHRANE (117). ** Automation tools were used; 228 records were excluded by a human, and 529 were excluded by automation tools applying filters (humans, clinical trial, age, and year of publication) Records were eliminated if they did not meet the inclusion criteria. Records removed for other reasons (e.g., language, ineligible reading). Studies not retrieved: text publications of articles that appeared to meet the eligibility. Registers: any additional records identified through reference lists.

**Figure 2 nutrients-14-04577-f002:**
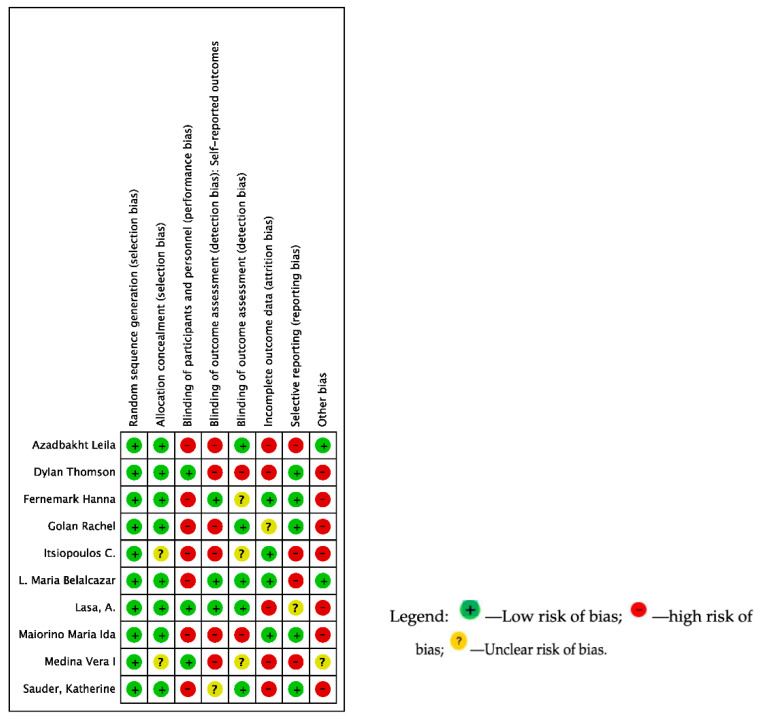
Risk of Bias of the Included Studies (n = 10).

**Figure 3 nutrients-14-04577-f003:**
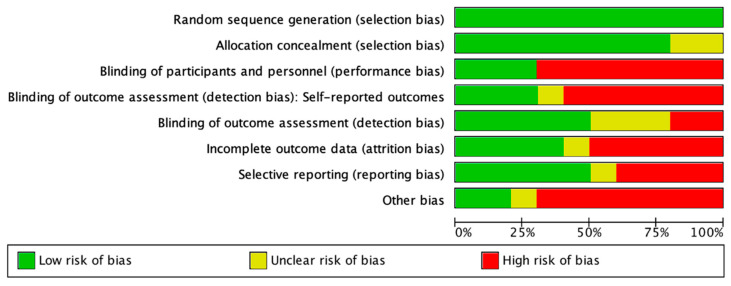
Risk of bias graph: authors’ judgements about each risk of bias item presented as percentages across all included studies (n = 10).

**Figure 4 nutrients-14-04577-f004:**
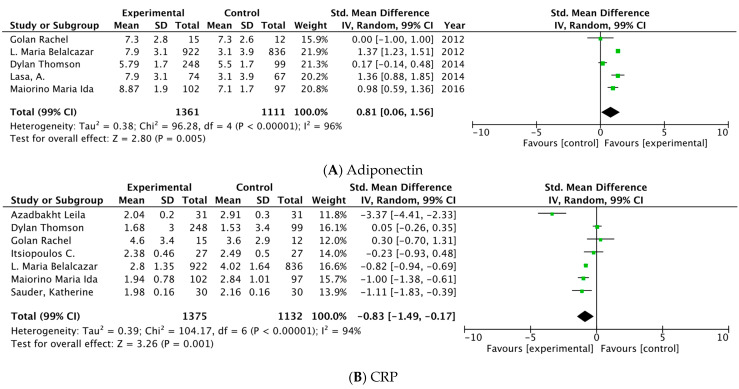
Meta-analysis of intervention studies. Forest plot of comparison between clinical trials presenting SMDS and 99% CI for DPs evaluated. Final values, outcome: (**A**) Adiponectin (ug/mL) with intake of Mediterranean Diet, Diabetes Prevention Program, Diabetes UK healthy eating guidelines. Outcome: (**B**) CRP (mg/L) with intake of Dietary Approaches to Stop Hypertension, Diabetes UK healthy eating guidelines, Mediterranean Diet, Diabetes Prevention Program, American Heart Association’s Therapeutic Lifestyle Changes diet. Abbreviations: CRP, C-Reactive Protein; WMD, weighted mean difference.

**Figure 5 nutrients-14-04577-f005:**
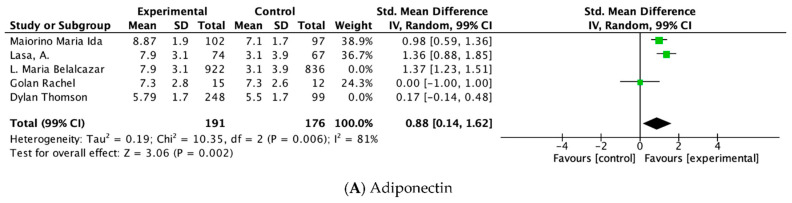
Subgroup by type of dietary pattern of intervention studies. Forest plot of comparison between clinical trials. Subgroup represents type of Dietary Pattern: MD is represented. Other studies with different DP are weighted 0%. Results present SMDS, and 99% CI for DPs evaluated. Final values, outcome: (**A**) Adiponectin (ug/mL0 and (**B**) CRP (mg/L) with the intake of Mediterranean Diet Abbreviations: CRP; C-Reactive Protein; WMD, weighted mean difference.

**Table 1 nutrients-14-04577-t001:** Summary table of studies included.

Author	Year	Study Name, Country	Follow-Up Duration	Number of Participants	Control/ Intervention Diet	Diet Assessment Method (Items)	Food and Average Components Intervention Diet	Glycemic Control Experimental vs. Control Groups	Biomarkers of Inflammation Experimental vs. Control Groups
**Thompson,** **2014** [35]	2014	United Kingdom ACTID	1 year	593 M: 385 F: 208	Standard dietary/Diabetes UK healthy eating guidelines	Food diaries	Diet was not prescriptive. Foods according to energy density and content of nutrients and glycemic index. Diet was oriented to goals.	HbA1c ^7^ 6.64 ^1^ (0.93) ^2^ Δ −27%** HOMA-IR ^9^ 1.60 ^1^ (0.74) ^2^ Δ −28%**	IL-6 ^a^ 1.85 ^1^ (−24, −0.9) ^5^ Δ −13% * CRP ^b^ 1.53 ^1^ (−27, −13) ^5^ Δ −9.2% * sICAM-1^c^ 232.3 ^1^ (−11, −3.1) ^5^ Δ −7.1% * Adiponectin ^d^ 5.79 ^1^, (−2.5, −14) ^5^ Δ + 5.2% *
**Maiorino, 2016** [32]	2016	Italy MÉDITA trial	8.1 years	215 M: 106 F: 109	Low fat diet/MD	Semi-quantitative FFQ Mediterranean Diet score	E%:1800 kcal/d (M), 1500 kcal/d (F), carbohydrate <50% of daily energy >30% fat calories. Plus: 30 to 50 g of olive oil.	HbA1c ^7^ 7.75 ^1^ ± 0.9 ^2^ Fasting glucose ^5^ 162 ^1^ (34) ^2^ HOMA-IR 5.2 ^1^ (1.7) ^2^ Δ 3.3 ^1^ ± 1.2 ^2,^*	CPR ^c^ 0.8^1^ (−1.3, −0.3) ^5^, Δ −37% Adiponectin ^d^ 1.9^1^ (0.8,3.09) * Δ + 43% *
**Lasa,** **2014** [34]	2014	Spain PREDIMED	1 year	191 M: 77 F: 114	Low-fat diet/MD	137-item FFQ	Subjects received intensive education to follow MD with qualitive aspects and quantitively: virgin olive oil (1 L/week) 30 g/d of mixed nuts (15 g walnuts, 7.5 g almonds, 7.5 g hazelnuts)) Positive and negative recommendations enriched: MUFA (50 mL) 1 L/week virgin olive oil (4 spoons oil/d)	HbA1c ^7^ 8.1 ^1^ (0.5) ^2^ Fasting glucose ^5^ 147.5 ^1^ (49.6) ^2^ Δ 3.6 (41.7) * HOMA-IR ^9^ 9.6 ^1^ (6.6) ^2^ Δ 0.4 ^1^ (4.8) ^2^	Adiponectin ^d^ +20.2 ^1^ (44.5) ^2,^* Visfatin ^c^ +1.2 ^1^ (5.5) ^2^ Leptin ^c^ −0.1 ^1^ (1.1) ^2^ TNF-α ^a^ + 1.7 ^1^ (23.3) ^2^ Resistin ^c^ −0.0 ^1^ (0.5) ^2^ A/L^d^ + 8.1 ^1^ (21.9) ^2,^* ICAM-a^c^ −110.5 ^1^ (10.1) ^2,^* IL-6 ^a^ −170.20 ^1^(8.3) ^2,^*
**Itsiopoulos,** **2011** [37]	2011	Australia	24 weeks* (12 weeks and then crossing over to the alternate diet)	27 M: 16 F: 11	Habitual diet/ MD	Self-completed seven-day diet record (in household measure)	E%: 11 MJ (2627 kcal) of total energy: medium carbohydrate (44% of energy, alcohol, 4% of energy from red wine), moderate protein (12% of energy), high in fat (40% of energy; > 50% from MUFA), olive oil (75 mL/d). Fiber (47 g/d), folate (700 mg/d), vitamin C (274 mg/d), carotenoids lycopene (14.4 mg/d), lutein/zeaxanthin (19.8 mg/d), fruits (563 g/day), vegetables (691 g/day) (280 g/day green leafy vegetables)	HbA1c ^7^ 6.8 ^1^ (6.3−7.3) ^5,**^ Fasting glucose ^5^ 8.9 ^1^ (7.8−10.0) ^5^ HOMA-IR ^9^ 5.2 ^1^(3.9−6.6) ^5^	CRP ^e^ CDt 2.49 ^1^, (1.69, 3.30) ^5^ MD 2.38 ^1^, (1.66, 3.10) ^5^ IL-6 ^a^ Δ −49% **
**Sauder,** **2015** [38]	2015	USA	8 weeks	30 M: 15 F: 15	American Heart Association’s Therapeutic Lifestyle Changes diet/+ pistachios	Daily compliance questionnaires	E%: Moderate energy fat (26.9%), SFA (6.7% of energy), and cholesterol (186 mg/d) + 20% of daily energy from pistachios.	HbA1c ^7^ 6.2 ^1^ (0.1) ^2^ Δ 6.0 ^1^ (0.0) ^2^ Fasting glucose ^5^ 106.29 ^1^ (10.81) ^2^ Δ 5.9 ^1^ (0.1) ^2^ HOMA-IR ^9^ 1.8 ^1^ (0.6) ^2^ Δ 1.6 ^1^ (0.1) ^2^	CRP ^e^ 1.98 ^1^ (0.16) ^2^ ICAM -1 ^c^ 112.1 ^1^ (5.8) ^2^ VCAM-1 ^c^ 337.7 ^1^ (21.2) ^2^ E-selectin ^c^ 47.1 ^1^ (3.6) ^2^
**Medina,** **2018** [40]	2018	Mexico	3 months	81 M: 46 F: 35	Placebo diet/ AD dietary portfolio	24 h dietary recall	E%: 45–55% carbohydrate, 15–20% protein, 25–35% fat (< 7% SFA), 200 mg/day cholesterol, 20–35 g of fiber and 2000–3000 mg/d of sodium Based on 200 kcal from diet: 14 g of dehydrated nopal, 30 g of soy protein, 4 g of chia seeds, and 4 g of inulin, + 15 g of maltodextrin and 28 g of calcium caseinate	HbA1c ^7^ 7.51 ^1^ (1.2) ^2^ Δ − 7.2% * Fasting glucose ^6^ 8.6 ^1^ (2.8) ^2^ Δ − 8.7% * HOMA-IR ^9^ 3.5 ^1^ (2.1) ^2^	CRP ^b^ Δ −13% * LPS ^e^ Δ −65% *
**Golan,** **2012** [33]	2012	Israel DIRECT trial	2 years	46 M: 42 F: 4	Low-fat diet/MD	FFQ	Moderate-fat, restricted calorie. Low energy diet E%: 1800 kcal/d (M), 1500 500 kcal/d (F), fat 35%; 30 to 45 g of olive oil + nuts (five to seven nuts, <20 g/d).	Fasting glucose ^5^ 142.8 ^1^ (53.08) ^2^ Δ − 7.72 (53.1) Fasting plasma insulin 14.2 ^1^ (10.05) Δ − 2.63 (5.4)	CRP ^b^ 5.0 ^1^ (3.4) ^2^ Δ − 0.66 ^1^ (3.0) ^2^ Leptin ^e^ M 7.8 ^1^ (4.4) ^2^ Δ − 1.1 ^1^ (2.8) ^2^ F 28.9 ^1^ (12.8) ^2^ Δ − 6.3 ± 6.8 Adiponectin ^e^ M 6.0 ^1^ (1.8) ^2^ Δ 0.5 ^1^ (1.9) ^2^ F 7.3 ^1^ (3.9) ^2^ Δ 1.5 ^1^ (2.5) ^2^
**Azadbakht,** **2011** [39]	2011	Iran	8 weeks	31 M: 13 F: 18	Control diet/DASH diet	3-D food diaries	Calculation of individual energy requirements. E%: 50–60% carbohydrates, 15–20% protein, <30% total fat, and <5% energy from simple sugars. High: vegetables, fruits, whole grains, and low-fat dairy products, minimum of saturated fat, cholesterol, refined grains, and sweets. Sodium 2400 mg/d +PUFA	Fasting glucose ^5^ Δ − 13.9 (4.5) **	hs-CRP ^b^ CDt 2.91 ^1^ (0.30)^2^ DASH 2.04 ^1^ (0.20)^2^* % Δ CDt − 5.1 ^1^ (3.8)^2^ DASH − 26.9 ^1^ (3.5)^2^ *
**Belalcazar, 2012** [36]	2012	USA Look AHEAD	1 year	1759 M: 720 F: 1039	Diabetes support and education/ DPP	FFQ	E%: < 114 kg: 1200––1500 kcal/d, ≥114 kg: 1500–1800800 kcal/d (low-calorie), fat diet (<3.030% of kcal. <10% SFA from fat), total energy: 1200 to 1800 kcal/d (>15% from protein and <30% of calories from fat). Meal replacement products: 1 portion-controlled snack, and 1 self-selected meal/day. At week 20: same prescribed meal replacement/d and two meals of self-selected foods. Continue dietary protocol for years 2–4 + 1 meal replacement/d.	HbA1c ^7^ 7.25 ^1^ (1.14) ^2^ Δ − 0.7 ^1^ (1.0) ^2,^* Fasting glucose ^5^ 152.19 ^1^ (44.71) ^2^ Δ − 21.7 ^1^ (44.4) ^2^ *	CRP ^b^ CDt 4.2 ^3^ (1.9,8.8) ^4^ DPP 4.2 ^3^ (1.9,9.1) ^4^ Δ − 1.24 (−3.4,0.01) * Adiponectin ^d^ CDt 4.8 ^3^(3.5, 7) ^4^ % Δ 0.2 ^3^ (−15.6, 20.1) ^4^ DPP 4.6 ^3^ (3.3, 6.6) ^4^ % Δ 11.9 (−7.2, 37.5) * HMW-adiponectin ^d^, CDt 1.9 ^3^ (1.2,3.2) ^4^ % Δ 0.9 ^3^ (−0.4,0.5) ^4^ DPP 1.9 ^3^ (1.1, 3.1) ^4^ % Δ 21.1 (−6.4,60.9) ^4,^*
**Fernemark,** **2013** [41]	2013	Sweden	9 weeks	19 M: 10 F: 9	Low-fat diet/MD	Directly report	E%: 1025–1080 kcal (M) and 905–984 kcal (F) 32–35% carbohydrates, protein 15%, fat 40%, MUFA 29.1 g, PUFA 8.3 g, SFA 8.1 g. (Not including food eaten at home later during the day) + 200 mL black coffee + red wine 14% alcohol (20 mL/150 mL) Ingested as one single large meal for lunch.	HbA1c ^8^ 51^1^ (10) ^2^ Fasting glucose ^6^ 80^1^ (17) ^2,^*	Leptin ^a^ CDt 16.758^1^ (11,611) ^2^ MD 13.822^1^(11,187) ^2,^*

Description of characteristics of studies included. Glycemic control and biomarkers of inflammation are presented as final values and changes. Values are presented as mean ^1^ and standard deviation ^2^ or median ^3^ and IQR ^4^ or 95% CI ^5^. Values representing glycemic control are represented by fasting glucose (mg/dL ^5^ or mmol/mol ^6^), HbA1c (% ^7^ or mmol/mol ^8^), Fasting plasma insulin (μU/mL). HOMA-IR ^9^: Homeostasis model assessment of insulin sensitivity = fasting insulin (μU/mL) fasting glucose (mmol/L)/22.5 and values >2.5 may indicate insulin resistance. Units of measure of inflammation biomarkers is presented in pg/mL ^a^, mg/L ^b^, ng/mL ^c^ μg/mL ^d^, mg/dL ^e^. Statistically significant difference: *p* < 0.001 *. *p* < 0.05 **. Abbreviations: ACTID, Early Activity in Diabetes; AD, anti-inflammatory diet; change, Δ; CRP, C-reactive protein; CI, confidence interval; DASH, Dietary Approaches to Stop Hypertension; d, day; CDt, control diet; DPP, Diabetes Prevention Program; DIRECT trial, Diabetes Remission Clinical Trial; E%, energy percent; F, female; HbA1c, glycated hemoglobin; HMW, High-Molecular-Weight Adiponectin; hs, high sensitivity; HOMA-IR, Homeostatic model assessment for insulin resistance; ICAM-1, intercellular adhesion molecule 1; IL-6, interleukin-6; IQR, interquartile range; LPS, lipopolysaccharide; Look AHEAD, Action for Health in Diabetes; M, male; MD, Mediterranean Diet; MÉDITA trial, Mediterranean Diet and type 2 diabetes; MUFA, mono-unsaturated fatty acid; PUFA, polyunsaturated fatty acid; PREDIMED, Prevención con Dieta Mediterránea (Prevention with the Mediterranean Diet); SFA, saturated fatty acid; UK, the United Kingdom; USA, the United States of America; VCAM-1, vascular, cell adhesion molecule-1; vs, versus.

**Table 2 nutrients-14-04577-t002:** Grading of Recommendations, Assessment, Development, and Evaluation (GRADE) evidence profile and summary of findings.

Certainty Assessment							No. of Patients		Effect		Certainty	Importance
No. of Studies	Study Design	Risk of Bias	Inconsistency	Indirectness	Imprecision	Other Considerations	Final Values	Diet Control	Relative (95%CI)	Absolute (95%CI)		
Adiponectin												
**5**	RCT	serious ^a^	very serious ^b^	NT	NT	Decrease the demonstrated effect by all plausible residual confounding	1361	1111	-	SMD **0.81** **higher** (0.06 higher to 1.56 higher)	**⊕⊕****◯◯** Low	Important
PCR												
**7**	RCT	serious ^a,b^	serious ^b^	NT	NT	The spurious effect is suggested by all plausible residual confounding, while no effect was observed	1375	1132	-	SMD **0.83** **lower** (1.49 lower to 0.17 lower)	**⊕⊕⊕****◯** Moderate	Important

Final values are compared to diet control for modulating chronic low-grade inflammation in patients with type 2-Diabetes Mellitus. Absolute effect is expressed in SMD (95%CI). a: Risk in do not blind personnel, patients and outcomes, b: meta-analysis, the I^2^ measure >90%. **⊕**: Ranking of certainty. Abbreviations: NT; not serious, RCT; randomized clinical trials. SMD: standardized mean difference.

**Table 3 nutrients-14-04577-t003:** Cluster of the Dietary Patterns and the most frequently used biomarkers.

Dietary Pattern	Adiponectin	IL-6	CRP	Leptin	I-CAM
Thompson, 2014 [35]	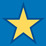	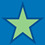	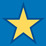	----	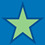
Belalcazar, 2012 [36]	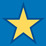	----	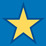	----	----
Medina, 2018 [40]	----	----	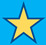	----	----
Maiorino, 2016 [32]	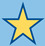	----	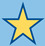	----	----
Lasa, 2014 [34]	----	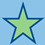	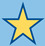	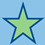	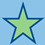
Itsiopulos, 2011 [37]	----	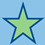	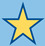	----	----
Fernemark, 2012 [41]	----	----	----	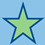	----
Golan, 2012 [33]	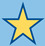	----	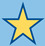	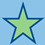	----
Sauder, 2015 [38]	----	----	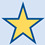	----	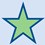
Azadbakht, 2012 [39]	----	----	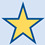	----	----

Studies were clustered by results, and similar features were summarized according to biomarkers of inflammation more frequently utilized in RCTs, grouping each DP. The CP cluster included recommendations by the American Heart Association and DASH diet, and the DE group included DPP and the Diabetes UK healthy eating guidelines. Adiponectin and CRP values marked in yellow color were used for meta-analysis. Other values marked in green color were not calculated in the meta-analysis due to insufficient studies. Abbreviations: AD 
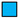
, anti-inflammatory diet; CRP, C-reactive protein; CP 
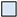
, Cardio Protective; DASH, Dietary Approaches to Stop Hypertension; ICAM-1, intercellular adhesion molecule; IL-6, interleukin-6; MD 
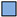
, Mediterranean Diet; DE 
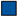
, specialized diet for subjects with diabetes.

## Data Availability

Data described in the manuscript will be made available upon request from the corresponding author.

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
