# Peer review of "The Effect of Dietary Patterns on Inflammatory Biomarkers in Adults with Type 2 Diabetes Mellitus: A Systematic Review and Meta-Analysis of Randomized Controlled Trials"

_nutrients, 2022, doi:10.3390/nu14214577_

Round 1

Reviewer 1 Report

Please see file attached. 

Author Response

Response to Reviewer Comments

1.  (Table 3) indicates that some of the studies reviewed also reported biomarkers such as leptin, IL-6, and I-CAM, but the results of these are never presented. Likewise, the same table (Table 3) claims that the studies reviewed evaluated DASH and other patterns, but few details are included in these studies reported, for example, in the abstract and conclusion. It is unclear if the authors focus on these aspects to report only ‘desirable’ outcomes or if, instead, they were only interested in the MD and CRP/adiponectin (authors, please clarify this in the manuscript)

Answer:  We showed in Table 1 all biomarkers of inflammation we found. Unfortunately, we tried diverse combinations in the meta-analysis, when not enough data were available for comparison between studies (we contacted the authors and received no answer) it was not possible to include them in our analysis, nevertheless we tried to show their presence in Table 3. Differences of means in glucose, HbA1c, and insulin are presented in table 1.

2.  The abstract and manuscript contain acronyms not described (MD, DP, RCT, CRP, PICO…), which would difficult understanding certain terms for some readers. Please define each acronym the first time it is used in both the abstract and main text. Similarly, please consider shortening sentences.

Answer:  We described and defined each acronym.

3.  Please also carefully check the grammar.

Answer:  We had the document checked for language by a professional service.

4.  The findings from the meta-analysis and systematic review conducted by the authors are barely mentioned. For example, it would be interesting to learn why they think there was a difference in adiponectin and CRP and discuss why this is important. Only then, the data should be compared to what other studies have found. Similarly, they mention that the American Diabetes Association suggests patterns to manage diabetes, but then they do not mention their research in relation to this literature; they only mention that DPs can decrease low-grade inflammation (line 361), but they do not use their findings to, for example, suggest which DP should be endorsed over others. The authors also mention that they compared different characteristics of foods shared in each DP, but then they do not link this information to their results. For example, do the foods shared in the DPs analysed differ a lot, thereby potentially affecting the effects of the meta-analysis, or were there similar foods included in each pattern? As a final example, the authors mention that a low-carbohydrate diet is recommended for people with diabetes. But then, this information is not mentioned in relation to their research. For example, does this imply that there should be a subgroup analysis where control groups are those with a low-carbohydrate diet?

Answer:  (Line 420) We extended our discussion section, according to the objectives of the study, dietary patterns, and their effects on biomarkers of inflammation. During the discussion, we approach topics about weight loss, bioactive compounds, and diverse comparisons in other reviews and studies.

Our main purpose when initiating this project was to explore opportunities to generate an anti-inflammatory dietary pattern specific for patients with type 2 diabetes mellitus.

We know that there are several systematic reviews of dietary patterns in Diabetes Mellitus, however, to our knowledge, there are none that explore the effect of dietary patterns on inflammatory biomarkers, and furthermore, we consider that our work contributes to suggesting the Mediterranean Diet as an anti-inflammatory pattern; it is important to continue understanding the synergy of nutrients, etiology, and treatment of the disease.

We believe that our findings will allow readers to understand the importance of identifying and controlling the onset of the inflammatory process in Diabetes Mellitus using biochemical markers and dietary patterns. Our study is a call to scientific communities to focus on homogenizing experimental diets, control diets, inflammatory markers and explore opportunities to generate an anti-inflammatory dietary pattern specific for patients with type 2 Diabetes Mellitus.

5.  Title: Specify that adults with T2DM are the population of interest. For example, The effect of anti-Inflammatory Dietary Patterns on adults with Type 2 Diabetes 2 Mellitus: A Systematic Review and Meta-Analysis of Randomized Controlled Trials

Answer:  We changed the title (line 1,2)

Abstract

6.  Line 14: the phrase “circulating markers of inflammation are especially applicable as descriptive on ongoing pathology” is confusing. Consider mentioning that markers of inflammation could indicate underlying ongoing pathology. Also, briefly provide examples of markers of inflammation.

Answer:  Line 14,15,16 we added this information

7.  Line 17: Specify that the objective is on participants with T2DM.

Answer:  Line 19 we specified this information

8.  Line 19: ‘mean values’ or ‘mean differences compared to the control group’?

Answer:  Mean differences

9.  Line 21: ‘intervention’ or ‘interventions’?

Answer:  Interventions

10.  Line: 21: This sentence implies that following any DP reduces CRP and increases adiponectin.

Answer:  Line 24, 25, we specificized this information

11.  Please specify which pattern(s) were studied.

Answer:  Line 25,26 we specified patterns studied.

12.  Line 22: Is not specified if a reduction in CRP and an increase in adiponectin are beneficial or not. Please add to this sentence that these values indicate more/less inflammation.

Answer:  We added this information (line 44)

13.  Line 24: the sentence starting with ‘subgroup analysis showed that…’ is incomplete.

Answer:  We completed the sentence.

14.  Line 34: authors claim that biomarkers of low-grade inflammation define components of T2DM; however, biomarkers of diabetes are typically glucose, glycated haemoglobin, etc.

Please, instead, mention that biomarkers of low-grade inflammation are relevant to the study of T2DM and explain why. For example, provide justification as to why it is relevant to focus on inflammation in individuals with T2DM. Is it because inflammation could lead to a more rapid progression of the disease?

Answer:  Line 37-37 We added and clarified low- grade biomarkers.

15.  Line 37: please explain in the manuscript why low-grade inflammation could indicate an ongoing pathology (ie, what do inflammation biomarkers indicate?).

Answer:  Line 41, we explained why low-grade inflammation could indicate an ongoing pathology

16.  Line 38: apologies, but I cannot understand this sentence, please re-phrase.

Answer:  We re-phased the sentence.

17.  Line 46: please briefly explain in the manuscript the biological mechanisms of how anti-inflammatory diet components could result in favourable changes in inflammatory biomarkers. Also, please provide examples of foods that contain antioxidants, prebiotics, mono and polyunsaturated fats.

Answer:  We explained in lines 50-51

18.  Line 48: provide examples of “bioactive compounds”.

Answer:  We explained in line 56

19.  Line 49: Please explain in the manuscript how a disbalance of inflammation would improve glycemic control? Ie, briefly explain the mechanism.

Answer:  We explained in line 60

20.  Line 55: Move the sentence starting on “as there as very few reviews that focus on T2DM…”at the end of the previous paragraph. It is useful for readers to know beforehand that there is limited research on the topic before knowing the purpose of the study.

Answer:  We moved the sentence to line 61.

21.  Line 68: Double-blind RCTs? Does this mean that neither patients nor researchers knew which diet was assigned to each patient? This is highly unlikely in nutrition research. Please explain in the manuscript how do these studies achieved a double-blind nutrition RCT.

Answer:  We did not restrict the search to “Double-blind RCT’s”, in the tool for Risk of Bias assessment, during analysis we assigned “low risk” if they were double blinded as part of performance. This has been corrected in the text.

22.  Line 72: confusing sentence, please rephrase.

Answer:  We rephased the sentence.

23.  Line 75: in the manuscript, please only provide examples of search terms. The complete search term of each electronic database should be provided as supplementary materials.

Answer:  We provided examples, line 83-86. We also provided some examples of search terms in the manuscript and added the complete search term of each electronic database in the Supplementary Materials.

24.  Line 94: please explain or re-phrase. How did the authors identify reports that did not appear through electronic searches? Does this mean that the authors consulted the references of studies identified through the search strategy?

Answer:  We explained in line 87.

25.  Line 97: the eligibility criteria do not mention aspects of the ‘comparison’. Ie, does not mention if only studies having a control group were included.

Answer:  We clarified in line 95.

26.  Line 99: what does “no minimum intervention mean?”, explain in the manuscript.

Answer:  We described it.

27.  Line 99, please explain the term ‘a priori’ to readers in the manuscript.

Answer:  We explained this term.

28.  Line 100: the authors mention twice that only studies in adults with diabetes were included, please only mention it once.

Answer:  We corrected this

29.  Line 100: “no minimum requirements for sample size” should not be part of the inclusion criteria. The inclusion criteria refer to the aspects that the authors specifically looked for.

While it is useful to mention this, please mention this after the inclusion criteria.

Answer:  We changed this phrase.

30.  Line 107: does this mean that changes in glucose, HbA1c, and insulin were also studied as a secondary outcome? I did not see a result of these outcomes.

Answer:  We added this in outcomes line 361.

31.  Line 108: please define in the manuscript ‘metabolic control’ of diabetes.

Answer:  We corrected this in line 102.

32.  Line 109: please justify in the manuscript why studies of a single mean would classify as those exploring dietary patterns, and define these types of studies (it is currently unclear what type of study would be this one).

Answer:  We justified this in the manuscript.

33.  Line 111: please explain the term ‘progress studies’. Perhaps authors mean ‘studies reporting on the progress of a previously implemented intervention’.

Answer:  We explained and added “studies reporting on the progress of a previously implemented intervention”

34.  Line 114: please explain which DPs were considered eligible. Was any study exploring any dietary pattern considered eligible? or only those that their respective authors described as an anti-inflammatory or healthy pattern? It is strange that only 10 RCTs were found if all DPs were eligible for inclusion. Besides, this information should be placed in the previous paragraph, as it is part of the inclusion criteria. Please use PICOS to explain eligibility criteria.

Answer:  We clarified aspects of PICOS and added this information.

35.  Line 124: the fact that the long-lasting follow-up was used for articles with information from the same study was already discussed in line 111. Also, the exclusion criteria should be in the ‘eligibility’ section, not in the ‘selection of studies’ section.

Answer:  We added this information in eligibility section. Line 90.

36.  Line 131: please explain what ‘if it was not possible to isolate effects of DP change’ means.

Answer:  We explained this phrase.

37.  Line 134: please use the term ‘duplicated’ articles instead of ‘repeated’ articles.

Answer:  We used duplicated’ articles

38.  Line 136: please explain

Answer:  We explained the meaning of a compendium of data was designed’.

39.  Line 138: what does ‘inclusion and exclusion criteria’ mean? Are authors referring to reasons to either include or exclude each article?

Answer:  yes, we were referring to reasons to either include or exclude each article, we clarify it.

40.  Line 138: Do authors mean ‘age and age range’?

Answer:  Yes, mean age and age range’. We corrected it.

41.  ‘Similarly, how do ‘gender’ and ‘gender distribution’ differ? Does ‘body mass index ‘refer to ‘mean body mass index’?

Answer:  Yes, body mass index ‘refer to ‘mean body mass index’. We corrected it.

42.  Line 139: “details of the intervention” such as? Please explain which details were extracted.

Answer:  We explained details, Line 130.

43.  Line 140: what does ‘inflammation parameter inflammatory with laboratory technique’ mean?

Answer:  We explained in line 131.

44.  Line 145: provide examples of “relevant declarations by the authors”.

Answer:  We provided examples.

45.  Line 147: please explain “clustering in accordance with the Cochrane handbook for systematic reviews of interventions”. How did the handbook suggest “clustering” the findings of this specific study?

Answer:  We provided examples in lines 139-140.

46.  Line 152: please re-phrase. Probably authors meant “disagreements were discussed to reach a consensus”. Also, please explain why disagreement resolution would be needed if only one person extracted the data (according to line 151).

Answer:  We explained and clarify this information.

47.  Line 158: the term PICO already includes ‘comparison’ and ‘outcome measurements’, no need to repeat these terms in the same sentence.

Answer:  We eliminated these terms.

48.  Line 159: “we made decisions regarding study limitations”. What does this mean?

Answer:  We explained in line 140.

49.  GRADEPro asks about the risk of bias, indirectness, etc to allocate a score?

Answer:  We explained in line 160.

50.  Line 167-171: please re-phrase to improve readability.

Answer:  We re-phrased it.

51.  Line 173: outcomes of interest were already mentioned in “eligibility criteria”. Please only mention it once in the manuscript.

Answer:  We eliminated it.

52.  Line 175: please explain how the authors “worked with the statistical properties of every dietary pattern” and how they “homogenized” the cut-off points. It is unclear what this means, especially if they were working with RCTs rather than observational studies, which are the ones that tend to use scores with cut-off points. Please also explain (in this letter, not in the manuscript) where the results of these “statistical properties” and homogenized” cut-off points are reported.

Answer:  We corrected and explained the results and terms “statistical properties” and “homogenized” “cut-off” points.

53.  Line 181: do authors mean “mean differences” instead of “mean values”?

Answer:  We used “mean differences”

54.  Line 182: what do authors refer to as “unifying data was required”?

Answer:  We explained it.

55.  Line 187: please re-phrase to improve readability.

Answer:  We re-phrase it,

56.  Line 187-189: not sure why studies with no “modification between values” were treated differently. Wouldn’t these studies simply have a mean difference of 0?

Answer:  We explained and corrected it.

57.  Line 193: why are DPs considered a confounder? This is the main exposure, not a covariate.

Answer:  We clarify confounders.

58.  Line 200: why are methods of the meta-analysis presented in line 180 and again in line 200?

Please present all methods related to a meta-analysis in one single section. Please ensure this is written in a way that is easy to understand for readers.

Answer:  We eliminated and synthetized information

59.  Line 210: why weren’t the standard deviation values asked to the original authors?

Answer:  We asked to the original authors

60.  Line 214: please add more details about the relevance of “converting between high- and low-sensitivity CRP”.

Answer:  We added why this is necessary.

61.  Line 215: were all subgroup analyses specified in the PROSPERO protocol registration?

Please state so in the manuscript. Also, provide an explanation of why so many subgroup analyses were carried out. Subgroup analysis should only be carried out when authors have a specific concern/theory in mind. For example, if authors consider that studies with longer durations have larger effects than those with shorter durations, etc.

Answer:  We stated this in this manuscript. Line 212.

Results

62.  Overall: The study claims to review 10 studies but only 7 are present in the meta-analysis.

Please clarify (in the flow diagram, abstract, and results) how many studies were used in the review and how many in the meta-analysis. Also, please clarify why not all studies were included in the meta-analysis.

Answer:  We clarified this information.

63.  Line 231: please provide details of this “dichotomous variable”. Is it a binary variable evaluating whether a person adheres or not to the MD?

Answer:  Yes, is binary variable. We corrected it.

64.  Line 234: regarding the study using insulin, does this refer to using insulin as medication? Or to use insulin values as the outcome of the intervention?

Answer:  is insulin medication,

65.  Line 234: what is an “incomplete monitoring assessment”? Explain in the manuscript.

Answer:  We explained it.

66.  Line 236: which specific information was not obtained after contacting the authors? (eg, standard errors?)

Answer:  We added that specific information

67.  Figure 1: the figure claims to exclude “records removed for other reasons”. Please provideexamples of these “other reasons” in either the figure or footnotes. Also, 228 records were excluded, but the figure does not explain why; is it because they did not meet the inclusion criteria? Explain the meaning of “6 studies sought for retrieval”.

Figure 1: In the section of reports excluded, be more specific. For example, write

‘participants without diabetes’ instead of ‘population’ or ‘not RCTs’ instead of ‘study design’.

Answer:  Figure 1. We corrected and clarified the PRISMA 2020 flow diagram and amended our systematic review record.

68.  Line 259: 47.8 + 51.21 does not equal 100%.

Answer:  We corrected it.

69.  Figure 2: the footnotes contain acronyms absent in the figure, please delete them. Also, this figure claims only to evaluate Mediterranean diets, but Table 3 claims to evaluate studies with the DASH diet also (among other types of diet). Please correct the mistake.

Answer:  We corrected this mistake. Figure 2 relates to Risk of Bias of the Included Studies and legends explain the symbols, we revised titles and footnotes.  

70.  Page 8: The second figure on this page does not have a title. This figure also contains sentences starting with a lower case, please correct this.

Answer:  We added title.

71.  Table 3: this table is missing headings. Also, there is no information on the characteristics of the sample (size, %females, etc), study design (eg, cross-over parallel design), intervention type (i.e., diet alone, diet + physical activity), etc.

Table 3: this table claims that studies also reported IL-6, Leptin, I-CAM. However, these findings are not discussed in the text; please add them.

Answer:  We added Table 1. With this information. In table 3, the most frequently used pattern was the Mediterranean diet, and we try to fit in characteristics because we noticed different approaches between studies.  Additionally, we reworded Table 3 and changed the name for a better explanation.

72.  Lines 288-298: Not all studies are described in this section. Tip: the details of each study could be included in a table only.

Answer:  We added Table 1. With this information.

73.  Table 4: the term “beneficial effect reported” should be substituted with “foods with beneficial effect reported” (the same for “not proven beneficial effect” which should be “foods with no beneficial effect reported”). Also, add a footnote to identify the source saying that these foods are beneficial. For example, are these considered beneficial/deleterious according to the WHO guidelines, the ADA, etc?

Table 4: this table contains sentences starting with lower case, please correct this.

Table 4: according to guidelines, not all drinks are “beneficial”, only non-caloric drinks.

Table 4: please edit this table to make it fit in 1-2 pages, it is difficult to read with this much blank space.

Answer:  We corrected Table 4.

74.  Line 300: this section seems to present the glycaemic control of patients at baseline and the duration of the diabetes diagnosis of participants. This information could be included in a section talking about the characteristics of the sample; please add a specific section discussing the characteristics of the participants. Sections discussing ‘outcomes’ should only include, for example, changes in blood glucose values in the studies evaluated, if any were found.

Answer:  Line 361, we corrected this information.

75.  Line 306: I find it concerning that “healthy subjects” were part of the control group and were compared to adults with diabetes. These are different samples that should not be compared as it introduces selection bias (unless there are individuals with and without diabetes in both the intervention and control groups). Moreover, this study should have been excluded during the screening phase, as the authors claimed to include only studies with participants with T2DM.

Answer:  We added this in discussion section.

76.  Line 339-348: the fact that lower heterogeneity in studies using the same type of diet is expected. Instead, it would be interesting to present if there were (or were not) differences in values of CRP or adiponectin according to subgroups.

Line 340: how was the MD associated with adiponectin and CRP? Ie, did it reduce or increase these values?

Answer:  We presented if there were differences in values of CRP or adiponectin according to subgroups. We integrated if MD was associated with adiponectin and CRP levels.

Discussion

77.  Line 425: what is the “standardised collection of nutritional, clinical and biomarker data”?

Answer:  We explained this.

78.  Line 425: What is meant by ‘joint elements to analyse’?

Answer:  We clarify how we do it.

79.  Line 431: why is a limitation that there are no specific cut-off values for chronic inflammation in T2DM? These can always be measured in continuous outcomes (i.e., areduction in CRP). Could this limitation be explained further?

Answer:  We explained this and added more details about this limitation.

80.  Line 437: since the “safety-related complications” are not mentioned before in the manuscript or thoroughly discussed, I suggest deleting this information.

Answer:  We deleted this information.

81.  Line 443, since the “insulin sensitivity-dependent and -independent immunomodulatory effects” are not previously mentioned or thoroughly discussed, please consider deleting this information or explaining it a bit further.

Answer:  We explained and corrected this information.

Conclusion

82.  Not sure how to provide feedback on the conclusion as there are several limitations in the study methods that hinder reaching one conclusion. For example, different DPs were evaluated, on different populations (individuals with and without diabetes), and only findings on CRP and adiponectin were discussed. Moreover, the authors claim that it is warranted to unify DPs to evaluate T2DM changes (line 457), but this is not possible. Dietary patterns widely differ in the literature; there are healthy, prudent, Western, traditional, DASH patterns etc., so it is impossible to ‘unify patterns’, they all have their own characteristics. It is possible to claim that Mediterranean diet definitions should be unified because this is only one pattern. However, the authors were not interested, nor they included MD patterns only. I encourage the authors to decide if the aim of the study was to focus on the study of the MD only or on comparing the effect of different DPs on inflammation-related outcomes because they appear to do the first one in the manuscript (eg, Table 2) but claim to do the latter one in their objectives.

Answer:  We re-wrote the conclusion to reflect the objectives of our systematic review and meta-analysis.

Other

83.  Line 475, financial support should be reported in funding, and please add the role of UAM/CONACYT in the study (generally, they do not take part in the study design, analysis or reporting of results).

Answer:  We corrected and clarify it.

Supplementary materials

84.  Please add table numbers to each table.

Pages 5 and 6 have numbers between paranthesis, please explain what this numbers mean in the tables (eg, SE or SD?).

Answer:  We added information required to the supplementary materials.

Reviewer 2 Report

Reviewer comments

Thank you for granting me the permission to the review this work. In this work, Sánchez-Rosales et al. conducted a systematic review and meta-analysis on the consumption of diets that demonstrate anti-inflammatory properties and its effects on the development of type 2 diabetes mellitus. Kindly, find below my comments for your response.

Abstract

Line 15: The authors should expand the abbreviation DP as it is the first time it has been introduced.

The authors should indicate the search “date” of the articles that were reviewed.

Line 16: replace “principal purpose” with “overarching aim”

Line 18: replace “operated by” with “conducted on”

Line 21: replace “significantly” with “significant”. Also, indicate the p-value that makes it significant

Line 22: bring “and” before the “plasma”

Line 24: This sentence “Subgroup analyses showed that when categorized by region”, is not complete. The authors should complete it

Keywords

Line 28: replace “Diabetes Mellitus type 2” with “Type 2 diabetes mellitus”

Introduction

Generally, the authors should consider splitting their long sentences.

Line 31-32: The authors should please split the sentence into two

Line 34-38: This sentence “Some biomarkers have been identified in the clinical evidence that demonstrate changes in low-grade inflammation, which define the components of T2DM; although this has not been well studied and provides interesting study potential Especially applicable is the fact that a state of chronic low-grade inflammation could be descriptive of ongoing pathology [3]” should be revised. The authors should split it into two as well. It is too long.

Line 43: Kindly, expand the abbreviation “DP” as it is newly used in the “Introduction”

Line 44: Kindly, split this sentence as it is too long “Nowadays, it is known that the relationship between inflammatory biomarkers and T2DM on disease progression, highlights the relevance of comprehension of the effects of Dietary Patterns (DP) on T2DM [8,9]; it is hypothesized that consumption of an anti-inflammatory diet (antioxidants, prebiotics, mono, and polyunsaturated fats, and bioactive compounds) would result in favorable changes on inflammatory biomarkers; every pattern reflects different combinations of food intake and they could contribute to this disbalance on inflammation, therefore to improve glycemic con-50 trol [10-12].”

Materials and Methods

Line 68: Why did the authors restrict themselves to “Double-blind RCTs” rather than “RCTs” so they rather assess the article quality using the Risk of Bias assessment which will address whether “Blinding” or no blinding was carried out?

Line 72: Revise this sentence “Search strategy was peer reviewed, advisory it was by a Scientometrics…..”

Line 75: Revise this “Search terms were included…” to “Search terms included were…………”

Line 114: revise “traditional” with “traditionally”

Line 116: revise “We considerate include dietary portfolios and eating…”

Line 120: This section “2.3. Selection of Studies” should rather come after “2.4. Data Extraction and methods for obtaining information”.

The authors have stated that they used “MENDELEY” as their Library Manager instead of using ENDNOTE. How were they able to sort out the DUPLICATE articles using MENDELEY.

Also, from the authors PRISMA guideline, they should show how many papers each were retrieved from each of the “Databases” used

Line 134: Do the authors mean “Duplicate articles” for the “repeated articles” they have stated?

Results

Line 191: Revise “I2” to “I2

Line 213: revise this “prioritized due to it is recognized function….”

Is Table 4 the summary Table? If yes, it is not clear and the authors have to thoroughly revise it. The summary Table should report the outcomes of all the final papers that were selected for the systematic review (from the PRISMA guideline). It should have the names of the authors, the country the studies were conducted, the characteristics of the participants, dosage of the intervention administered and the control or placebo treatment and results/findings. This information is important as it is what the authors should discuss.

Fig. 1. The PRISMA guideline presented is not correct and has to be revised. For example, the authors have indicated in the box “Studies sought for retrieval” as n=6. Yet, in the next box following it indicated “Reports assessed for eligibility” n=19. How can the n=6 papers yield n=19? This is wrong.

Discussion

The work is not properly discussed. This is partly attributed to how the Table of the selected papers from the PRISMA guideline is selected. The Table summary should have Geographical location of where the studies are coming from. The authors could have discussed that. Were the studies reported coming from the Mediterranean region?

Conclusion

The authors should re-write the conclusion to reflect the “Objectives” of this systematic review and meta-analysis.

General comments

The authors indicate “Anti-inflammatory Dietary patterns” in the Title. They must therefore define what their reference to “Anti-inflammatory Diet” means. Would the DASH diet too fall under “Anti-inflammatory Diet?”

From the authors search terms, I only see “Mediterranean Diet”. How come the authors didn’t use a “Plant-based diet” as one of the terms as they are prominent anti-inflammatory sources. From Line 114-115, the definition of “DP” stated by the authors does not connote the impression of “Anti-inflammatory Diet”.

Also, I expect that, in the Table of summary, the authors indicate the “Dosage of bioactive compounds” in the diets selected as that is what demonstrate the “anti-inflammatory activity”.

Generally, the manuscript needs thorough re-writing and proof-reading.

Author Response

Response to Reviewer Comments

Abstract

1.  Line 15: The authors should expand the abbreviation DP as it is the first time it has been introduced.

Answer:  We expanded the abbreviation

2.  The authors should indicate the search “date” of the articles that were reviewed.

Answer:  This has been indicated

3.  Line 16: replace “principal purpose” with “overarching aim”

     Line 18: replace “operated by” with “conducted on”

     Line 21: replace “significantly” with “significant”. Also, indicate the p-value that makes it significant

Answer:  Lines 16, 18 and 21 we rectified these terms.

4.  Line 22: bring “and” before the “plasma”

Answer:  Corrected

5.  Line 24: This sentence “Subgroup analyses showed that when categorized by region”, is not complete. The authors should complete it

Answer:  Corrected

Keywords

6.  Line 28: replace “Diabetes Mellitus type 2” with “Type 2 diabetes mellitus”

Answer:  We introduced all terms and words suggested, replaced, or eliminated them (line 15, 16, 18, 21, 22, 24, 28).

Introduction

7.  Generally, the authors should consider splitting their long sentences.

Answer:  We divided our long sentences for a better understanding and revised them.

8.  Line 31-32: The authors should please split the sentence into two

Answer:  Line 31-32, we split the sentence into two.

9.  Line 34-38: This sentence “Some biomarkers have been identified in the clinical evidence that demonstrate changes in low-grade inflammation, which define the components of T2DM; although this has not been well studied and provides interesting study potential Especially applicable is the fact that a state of chronic low-grade inflammation could be descriptive of ongoing pathology [3]” should be revised. The authors should split it into two as well. It is too long.

Answer:  Line 34-38, Line 31-32, we split the sentence into two.

10.  Line 43: Kindly, expand the abbreviation “DP” as it is newly used in the “Introduction”

Answer:  We expended the abbreviation.

11.  Line 44: Kindly, split this sentence as it is too long “Nowadays, it is known that the relationship between inflammatory biomarkers and T2DM on disease progression, highlights the relevance of comprehension of the effects of Dietary Patterns (DP) on T2DM [8,9]; it is hypothesized that consumption of an anti-inflammatory diet (antioxidants, prebiotics, mono, and polyunsaturated fats, and bioactive compounds) would result in favorable changes on inflammatory biomarkers; every pattern reflects different combinations of food intake and they could contribute to this disbalance on inflammation, therefore to improve glycemic con-50 trol [10-12].”

Answer:  Line 44, we split the sentence into two.

Materials and Methods

12.  Line 68: Why did the authors restrict themselves to “Double-blind RCTs” rather than “RCTs” so they rather assess the article quality using the Risk of Bias assessment which will address whether “Blinding” or no blinding was carried out?

Answer:  We did not restrict the search to “Double-blind RCT’s”, in the tool for Risk of Bias assessment, during analysis we assigned “low risk” if they were double blinded as part of performance. This has been corrected in the text.

13.  Line 72: Revise this sentence “Search strategy was peer reviewed, advisory it was by a Scientometrics…..”

       Line 75: Revise this “Search terms were included…” to “Search terms included were…………”

       Line 114: revise “traditional” with “traditionally”

       Line 116: revise “We considerate include dietary portfolios and eating…”

Answer:  Line 72, 75, 114, 116. We corrected these sentences and terms.

14.  Line 120: This section “2.3. Selection of Studies” should rather come after “2.4. Data Extraction and methods for obtaining information”.

Answer:  We changed sections 2.3 and 2.4. We had the document checked for language by a professional service.

15.  The authors have stated that they used “MENDELEY” as their Library Manager instead of using ENDNOTE. How were they able to sort out the DUPLICATE articles using MENDELEY.

Answer:  We revised manually, and we stated this in the manuscript.

16.  Also, from the authors PRISMA guideline, they should show how many papers each were retrieved from each of the “Databases” used

Fig. 1. The PRISMA guideline presented is not correct and has to be revised. For example, the authors have indicated in the box “Studies sought for retrieval” as n=6. Yet, in the next box following it indicated “Reports assessed for eligibility” n=19. How can the n=6 papers yield n=19? This is wrong

Answer:  We corrected and clarified the PRISMA 2020 flow diagram and amended our systematic review record.

17.  Line 134: Do the authors mean “Duplicate articles” for the “repeated articles” they have stated?

Answer:  Yes, we use “repeated articles”. We corrected it.

18.  Line 191: Revise “I2” to “I2

       Line 213: revise this “prioritized due to it is recognized function….”

Answer:  Line 191, 213, we revised and corrected it.

19.  Is Table 4 the summary Table? If yes, it is not clear and the authors have to thoroughly revise it. The summary Table should report the outcomes of all the final papers that were selected for the systematic review (from the PRISMA guideline). It should have the names of the authors, the country the studies were conducted, the characteristics of the participants, dosage of the intervention administered and the control or placebo treatment and results/findings. This information is important as it is what the authors should discuss.

Answer:  Table 4 is not the summary; it is a qualitative comparison of differences between Mediterranean Patterns we found. According to the PRISMA guideline, Table 1. is our summary table. We replaced Table 1, indicating dosage of bioactive compounds, and the diet selection as that is what demonstrates the “anti-inflammatory activity”.

Discussion

20.  The work is not properly discussed. This is partly attributed to how the Table of the selected papers from the PRISMA guideline is selected. The Table summary should have Geographical location of where the studies are coming from. The authors could have discussed that. Were the studies reported coming from the Mediterranean region?

Answer:  We extended our discussion section, according to the objectives of the study, dietary patterns, and their effects on biomarkers of inflammation. During the discussion, we approach topics about weight loss, bioactive compounds, and diverse comparisons in other reviews and studies. And discuss geographical location, and added the studies reported coming from Mediterranean region.

Conclusion

21.  The authors should re-write the conclusion to reflect the “Objectives” of this systematic review and meta-analysis.

Answer:  We re-wrote the conclusion to reflect the objectives of our systematic review and meta-analysis.

We know that there are several systematic reviews of dietary patterns in Diabetes Mellitus, however, to our knowledge, there are none that explore the effect of dietary patterns on inflammatory biomarkers, and furthermore, we consider that our work contributes to suggesting the Mediterranean Diet as an anti-inflammatory pattern; it is important to continue understanding the synergy of nutrients, etiology, and treatment of the disease.

We believe that our findings will allow readers to understand the importance of identifying and controlling the onset of the inflammatory process in Diabetes Mellitus using biochemical markers and dietary patterns. Our study is a call to scientific communities to focus on homogenizing experimental diets, control diets, inflammatory markers and explore opportunities to generate an anti-inflammatory dietary pattern specific for patients with type 2 Diabetes Mellitus.

General comments

22.  The authors indicate “Anti-inflammatory Dietary patterns” in the Title. They must therefore define what their reference to “Anti-inflammatory Diet” means. Would the DASH diet too fall under “Anti-inflammatory Diet?”

Answer:  We defined our reference for “Anti-inflammatory Diet” in Materials and methods, and we extended this definition.

23.  From the authors search terms, I only see “Mediterranean Diet”. How come the authors didn’t use a “Plant-based diet” as one of the terms as they are prominent anti-inflammatory sources. From Line 114-115, the definition of “DP” stated by the authors does not connote the impression of “Anti-inflammatory Diet”.

Answer:  In our search strategy, we included terms and mesh terms with Dietary Patterns and we found some articles with Plant-based diet (search terms: (Anti-inflammatory* Diet*” OR “score” OR “pattern” OR “adherence” OR “index” OR diet* index* OR dietary* patterns*” OR “ eating pattern” OR “eating* patterns*” OR “food* pattern*” OR “food* patterns*” OR “dietary* habit*” OR “feeding* behavior*”,“dietary habit”, “feeding behavior”). In the screening phase, we found some vegetarian styles and we excluded some of them according to our purpose and definition about anti-inflammatory patterns, these articles were not included in our study (eligibility criteria and outcomes of interest).

24.  With respect to Vegetarian styles or "diet, vegetarian" [MeSH Terms] OR Plant-based diet [Text Word], in a preliminary search we did not find RCTs with impact on biomarkers of chronic inflammation, we included this in the limitations section, and we pretend to update our systematic review.

Also, I expect that, in the Table of summary, the authors indicate the “Dosage of bioactive compounds” in the diets selected as that is what demonstrate the “anti-inflammatory activity”.

Answer:  We inserted Table 1, with these characteristics.

25.  Generally, the manuscript needs thorough re-writing and proof-reading.

Answer:  We had the document checked for language by a professional service.

Reviewer 3 Report

Aim of this systematic review and meta-analysis is to explore the inflammatory effect of some dietary patterns in T2DM.

The work is carried out in a very accurate and methodologically correct manner. 

MINOR POINTS

1. There are some acronyms in the abstract to be defined as DP and MD

2. Table 3 "Characteristics of food and average components among studies” is unclear. It needs to be reworded.

3. Table 4  "Characteristics and properties of components on Mediterranean Diet styles." is difficult to read. It must be placed horizontally.

4. The authors should deepen the discussion because it appears sterile. What are the characteristic foods of the Mediterranean diet that would benefit the degree of inflammation in diabetic patients? What is the role of alcohol?

It is also important to consider weight loss. What are the effects of the Mediterranean diet on weight loss in diabetics? What are the effects of Mediterranean diet-induced weight loss on visceral adipose tissue?

Author Response

Response to Reviewer Comment

1.  There are some acronyms in the abstract to be defined as DP and MD

Answer:  We defined and clarified terms; “DP” as Dietary patterns and “MD” as Mediterranean Diet. We had the document checked for language by a professional service.

2.  Table 3 "Characteristics of food and average components among studies” is unclear. It needs to be reworded.

Answer:  We rephrased Table 3 and changed the name for a better comprehension.

3.  Table 4 "Characteristics and properties of components on Mediterranean Diet styles." is difficult to read. It must be placed horizontally.

Answer:  Table 4 was placed horizontally.

4.  The authors should deepen the discussion because it appears sterile. What are the characteristic foods of the Mediterranean diet that would benefit the degree of inflammation in diabetic patients? What is the role of alcohol?

It is also important to consider weight loss. What are the effects of the Mediterranean diet on weight loss in diabetics? What are the effects of Mediterranean diet-induced weight loss on visceral adipose tissue?

Answer:  We discussed characteristics of foods in the Mediterranean Diet, and we extended foods and compounds as alcohol.

Mainly, we extended our discussion section, according to the objectives of the study, dietary patterns, and their effects on biomarkers of inflammation. During the discussion, we approach topics about weight loss, bioactive compounds, and diverse comparisons in other reviews and studies.

Our main purpose when initiating this project was to explore opportunities to generate an anti-inflammatory dietary pattern specific for patients with type 2 diabetes mellitus.

We know that there are several systematic reviews of dietary patterns in Diabetes Mellitus, however, to our knowledge, there are none that explore the effect of dietary patterns on inflammatory biomarkers, and furthermore, we consider that our work contributes to suggesting the Mediterranean Diet as an anti-inflammatory pattern; it is important to continue understanding the synergy of nutrients, etiology, and treatment of the disease.

We believe that our findings will allow readers to understand the importance of identifying and controlling the onset of the inflammatory process in Diabetes Mellitus using biochemical markers and dietary patterns. Our study is a call to scientific communities to focus on homogenizing experimental diets, control diets, inflammatory markers and explore opportunities to generate an anti-inflammatory dietary pattern specific for patients with type 2 Diabetes Mellitus.

Round 2

Reviewer 1 Report

The authors have addressed some of my minor comments, but my main concerns remain, see attached. 

Reviewer 2 Report

Thank you for making time to revise the manuscript.

There are some other minor revisions required.

Line 15: Kindly, revise the sentence  “……dietary patterns (DPs)Patterns……”

Line 17:  The aim “…………to explore the inflammatory effect of some DPs on participants with T2DM” stated is ambiguous. I thought the aim was rather to investigate the anti-inflammatory properties associated with adhering to certain dietary patterns?

Line 18: Do the authors mean they searched the databases last year up to this year? It is not clear. In systematic reviews, it is important the search process is carried out on a particular day. This allows for easy repeatability and reproducibility of the review.

Line 20: Please, revise the sentence “A total of 8 RCTs were included Meta-analysis;…….”

Line 26: There is a duplicate of I2. The authors should delete one of it.

Line 81: The authors should be specific with the search terms used and not indicate “examples” of some of the search terms. This is important to ensure repeatability of the review.
